# Machilin D, a Lignin Derived from *Saururus chinensis*, Suppresses Breast Cancer Stem Cells and Inhibits NF-κB Signaling

**DOI:** 10.3390/biom10020245

**Published:** 2020-02-05

**Authors:** Xing Zhen, Hack Sun Choi, Ji-Hyang Kim, Su-Lim Kim, Ren Liu, Bong-Sik Yun, Dong-Sun Lee

**Affiliations:** 1Interdisciplinary Graduate Program in Advanced Convergence Technology and Science, Jeju National University, Jeju 63243, Korea; zhenxing19932013@gmail.com (X.Z.); seogwi12@naver.com (J.-H.K.); ksl1101@naver.com (S.-L.K.); liuren0308@gmail.com (R.L.); 2Subtropical/Tropical Organism Gene Bank, Jeju National University, Jeju 63243, Korea; choix074@jejunu.ac.kr; 3Division of Biotechnology, College of Environmental and Bioresource Sciences, Jeonbuk National University, Gobong-ro 79, Iksan 54596, Korea; bsyun@jbnu.ac.kr; 4Practical Translational Research Center, Jeju National University, Jeju 63243, Korea; 5Faculty of Biotechnology, College of Applied Life Sciences, Jeju National University, SARI, Jeju 63243, Korea

**Keywords:** breast cancer stem cells (BCSCs), mammospheres, machilin D, NF-κB

## Abstract

Cancer stem cells are responsible for breast cancer initiation, metastasis, and relapse. Targeting breast cancer stem cells (BCSCs) using phytochemicals is a good strategy for the treatment of cancer. A silica gel, a reversed-phase C18 column (ODS), a Sephadex LH-20 gel, thin layer chromatography, and high-performance liquid chromatography (HPLC) were used for compound isolation from *Saururus chinensis* extracts. The isolated compound was identified as machilin D by mass spectrometry and nuclear magnetic resonance (NMR). Machilin D inhibited the growth and mammosphere formation of breast cancer cells and inhibited tumor growth in a xenograft mouse model. Machilin D reduced the proportions of CD44^+^/CD24^-^ and aldehyde dehydrogenase 1 (ALDH1)-positive cells. Furthermore, this compound reduced the nuclear localization of the NF-κB protein and decreased the IL-6 and IL-8 secretion in mammospheres. These results suggest that machilin D blocks IL-6 and IL-8 signaling and induces CSC death and thus may be a potential agent targeting BCSCs.

## 1. Introduction

*Saururus chinensis* is a medicinal perennial herbaceous plant that is mainly distributed in moist and wet locations in Japan, southern Korea, North America and China, and has been used in traditional medicine and resources to treat several diseases [1,2,3]. In cancer chemotherapy, synthetic anticancer agents are effective, but the repeated use of these agents in a complex tumor microenvironment often results in drug resistance [4]. Bioactive chemicals from *S. chinensis* have received increased attention as an alternative source of materials for cancer therapy. Several compounds, such as lignans, diterpenes, alkaloids, tannins, flavonoids, steroids, and lipids, isolated from *S. chinensis* possess a wide array of pharmacological and biochemical activities [5,6], such as antioxidant [7], antidiabetic [8], anti-inflammatory [1] and anticancer [9] activities.

Breast cancer is one of the most lethal malignant adenocarcinomas and a major cause of cancer-related death in women [10]. Globally, 15%–20% of female breast cancer patients are diagnosed with triple negative breast cancer (TNBC) based on the expression of the estrogen receptor, progesterone receptor, and epidermal growth factor receptor 2 [11].

TNBC is characterized by a high risk of recurrence, metastasis, and short progression-free survival (PFS) [12,13]. In recent decades, TNBC cells have shown to have properties similar to breast cancer stem cells (BCSC), and strategies targeting CSCs have shown therapeutic efficacy in preclinical studies of TNBC [14]. CSCs, a subpopulation of tumor cells, are cancer stem-like cells [15]. CSCs can promote oncogenesis to form the tumor bulk, including that of breast cancer, through self-renewal and differentiation [16]. In cancer chemotherapy and radiotherapy, CSCs show multidrug resistance and radio resistance, resulting in cancer recurrence and metastasis [17,18]. Therefore, targeting CSCs in cancer therapies is important.

Biomarkers of BCSCs, including CD44 and aldehyde dehydrogenase 1 (ALDH1), can be regulated during cancer progression and metastasis [19]. In TNBC patients, CD44 promotes the transcription of PD-L1, an immune checkpoint, through its cleaved intracytoplasmic domain (ICD) [20]. Inhibition of ALDH1 in breast cancer by curcumin decreased multidrug resistance [21]. The Wnt, Hedgehog, Stat3, Hippo, Notch, and NF-κB signaling pathways regulate CSC stemness and differentiation. Inhibition of BCSCs through targeting these molecular pathways can be an effective tool for cancer therapy [22,23]. Stem cell factors such as Sox2 and c-Myc are essential for BCSCs [24]. In the tumor microenvironment, cytokines such as IL-6 regulate the interaction between CSCs and cancer cells. Stat3 and NF-κB signaling stimulates IL-6 and IL-8 production to drive CSC formation [25].

Recently, *S. chinensis* extracts have been applied to various cancer cell lines, including gastric cancer [9], renal cell carcinoma [26], and hepatocellular carcinoma cell lines [27]. However, no reports have shown the effects of machilin D, a lignin obtained from *S. chinensis* extracts, on CSC formation. In our study, we purified machilin D from *S. chinensis* and showed that it suppressed the formation of CSCs. We demonstrated that machilin D inhibits BCSC activity through regulation of IL-6 and IL-8.

## 2. Materials and Methods

### 2.1. Reagents

Open column chromatography was performed using silica gel 60 A (Analtech, Newark, DE) and Sephadex LH 20 (Pharmacia, Uppsala, Sweden). Thin-layer chromatography (TLC) was carried out using a silica gel Kieselgel 60 F_254_ plate (Merck, Darmstadt, Germany). Preparative high-performance liquid chromatography (HPLC) was conducted on a Shimadzu system (Kyoto, Japan). Machilin D was obtained from the National Institute for Korean Medicine Development (Gyeongsan, Korea). The other chemicals were purchased from Sigma-Aldrich (St. Louis, MO, USA).

### 2.2. Plant Material

*S. chinensis* was purchased from Handsherb (Yeongcheon, Korea). The voucher specimen (No. 2017_020) is managed in the Department of Biotechnology, Jeju National University, JeJu, South Korea.

### 2.3. Extraction and Isolation

Dry powder of *S. chinensis* was extracted with methanol. The bioassay-based isolation protocol is summarized in Figure 1A. The extracts were vacuum-dried, and the sample was solubilized with 1000 mL of methanol. The methanol extracts were mixed with water, and the methanol was evaporated. The water-suspended samples were extracted with the same volume of ethyl acetate. The concentrated sample was loaded onto a silica gel column (3 × 35 cm) and fractionated with solvent (chloroform-methanol, 30:1) (Appendix A). The twelve parts were divided and assayed for mammosphere formation. The #5, #6, #7, and #8 fractions potentially inhibited mammosphere formation. The #5, #6, #7, and #8 fractions were subjected to preparatory C-18 open columns (5 × 7 cm) and eluted with 30%, 50%, 70%, and 100% acetonitrile (Appendix A). Four fractions were obtained and assayed for mammosphere formation. The 50% acetonitrile-eluted fraction inhibited mammosphere formation. The 50% acetonitrile fraction was loaded onto a Sephadex LH 20 open column (2.5 × 30 cm) and eluted with four fractions (Appendix A). The four fractions were obtained and assayed for mammosphere formation. Fractions #1 and #2 inhibited mammosphere formation. Fractions #1 and #2 were isolated using preparatory TLC (glass plate; 20 × 20 cm) and developed in a TLC glass chamber. Individual bands were separated from the silica gel plates using a surgery knife and collected in 15 mL conical tubes. Each fraction was obtained and assayed for mammosphere formation (Appendix A). Active parts #4 and #5 were subjected to HPLC. HPLC analysis was performed using a Shimadzu HPLC 20A system (Shimadzu, Tokyo, Japan). HPLC separation was conducted using an ODS 10 × 250 mm C18 column (flow rate; 3 mL/min). The mobile phase was composed of water (solvent A) and acetonitrile (solvent B). For gradient elution, solvent B was initially set at 20%, increased to 80% at 20 min and increased to 100% at 40 min. The purified sample was detected at a retention time of 14 min (Appendix A).

### 2.4. Structural Analysis of the Purified Sample

The chemical structures of the isolated compounds were determined by mass spectrometry and NMR measurements. The molecular weight was established as 344 kDa by ESI mass spectrometry, which showed quasi-molecular ion peaks at *m/z* 367.2 [M+Na]^+^ in positive mode (Appendix A). The ^1^H NMR spectrum in CD_3_OD exhibited signals due to six aromatic methine protons at δ 6.99, 6.97, 6.90, 6.83, 6.83, and 6.76, which are attributable to two 1,2,4-trisubstituted benzenes, two *trans*-conjugated olefinic methine protons at δ 6.32 and 6.14, two oxygenated methine protons at δ 4.63 and 4.37, two methoxy groups at δ 3.84 and 3.83, and two methyl groups at δ 1.83 and 1.05. In the ^13^C NMR spectrum, 20 carbon peaks included four oxygenated sp^2^ quaternary carbons at δ 151.8, 148.8, 147.8, and 147.3; eight sp^2^ methine carbons at δ 131.9, 124.9, 121.2, 120.0, 118.2, 115.8, 111.9, and 110.9; two sp^2^ quaternary carbons at δ 133.9 and 133.7; two oxygenated methine carbons at δ 81.9 and 78.3; two methoxy carbons at δ 56.5 and 56.4; and two methyl carbons at δ 18.5 and 16.5 (see Appendix A). All proton-bearing carbons were assigned by the HMQC spectrum, and the ^1^H-^1^H COSY spectrum revealed four partial structures (see Appendix A). Further structural elucidation was performed with the aid of the HMBC spectrum, which showed long-range correlations from the methine proton at δ 4.63 to the carbons at δ 133.7, 121.2, and 111.9; from the methine proton at δ 6.32 to the carbons at δ 133.9, 120.0, and 110.9; and from the methine proton at δ 4.37 to the carbons at δ 147.8 and 133.7. Finally, two methoxy groups were connected by the long-range correlations from the methyl protons at δ 3.84 to the carbon at δ 151.8 and from the methyl protons at δ 3.83 to the carbon at δ 148.8 (see Appendix A). Therefore, the structure of the isolated compound was identified as machilin D (Figure 2).

### 2.5. Cell Culture and Mammosphere Formation

Two breast cancer cell lines, MCF-7 and MDA-MB-231, were obtained from the American Type Culture Collection (Rockville, MD, USA). All human breast cancer cells were maintained in DMEM with 10% fetal bovine serum (HyClone Fisher Scientific, CA, USA) and 1% penicillin/streptomycin (HyClone, Thermo Fisher Scientific, CA, USA). Cancer cells (3.5 × 10^4^ or 0.5 × 10^4^ cells) were cultured in an ultralow attachment 6-well plate with a MammoCult^TM^ culture medium (StemCell Technologies, Vancouver, BC, Canada). All cells were maintained in a humidified 5% CO_2_ incubator at 37 °C for seven days. The mammosphere formation was quantified by the NICE program [28]. Mammosphere formation was estimated by examining the mammosphere formation efficiency (MFE) (%) [29].

### 2.6. Cell Growth Assay

MDA-MB-231 and MCF-7 cells (1 × 10^4^ cells/well) were seeded in a 96-well plate for 24 h and treated with various concentrations (10, 25, 50, 100, and 200 μM) of machilin D for 24 h in a culture medium. Then, growth assay was assessed using the EZ-Cytox kit (DoGenBio, Seoul, Korea) and the OD at a wavelength of 450 nm was measured using a GloMax^®^ Explorer Multimode microplate reader (Promega, Madison, WI, USA).

### 2.7. Colony Formation Assay

MDA-MB-231 cells (1 × 10^3^ cells/well) were seeded in a 6-well plate, treated with different concentrations of machilin D in DMEM and maintained for seven days at 37 °C in a 5% CO_2_ incubator. The grown colonies were washed with 1 × PBS three times, fixed for 10 min using 3.7% formaldehyde, treated for 20 min with 100% methanol, and stained for 30 min with 0.05% crystal violet. The colony formation plate was washed three times using 1 × PBS prior to image capture.

### 2.8. Wound-Healing Assay

MDA-MB-231 cells were plated in a 6-well plate at 1 × 10^6^ cells/well. The cells were cultured overnight and grown into a monolayer. A linear scratch was made by using an SPL Scar^TM^ scratcher (SPL Life Science, Pocheon, Korea). After the cells were washed three times with 1 × PBS, the breast cancer cells were treated with machilin D in a new DMEM. Images of the wounded areas were captured using a light microscope after 24 h.

### 2.9. Transwell Assay

Invasion and migration assays were performed in 12-well hanging inserts with pore polycarbonate membranes (Merck Millipore, Darmstadt, Germany) coated with (invasion) or without (migration) a growth factor-reduced Matrigel matrix basement (BD, San Jose, CA, USA) following the manufacturer’s protocol. Two hundred microliters of MDA-MB-231 cell suspensions in 1% FBS DMEM was added to the upper chamber (1 × 10^5^ cells/chamber). The bottom chamber was filled with 800 μL of DMEM containing 10% FBS as a chemoattractant. The cells were maintained for two days at 37 °C in a 5% CO_2_ incubator. The cells that passed through the membrane to the lower surface were fixed with 3.7% paraformaldehyde and stained with 0.05% crystal violet. The images were captured with a light microscope.

### 2.10. Flow Cytometric Analysis and ALDH1 Activity

After incubation with machilin D for 24 h, the cancer cells were harvested and dissociated using 1 × trypsin/EDTA. We used a previously described method [24]. A total of 1 × 10^6^ cells were cultured with anti-CD44-FITC and anti-CD24-PE antibodies (BD, San Jose, CA, USA) on ice for 30 min. The cancer cells were centrifuged and washed three times with a 1 × FACS buffer and analyzed by an Accuri C6 cytometer (BD, San Jose, CA, USA). The ALDH1 activity was assayed using an Aldefluor^TM^ assay kit (StemCell Technologies, Vancouver, BC, Canada). We used a previously described method [24]. The breast cancer cells were treated with machilin D (50 µM) for 24 h and reacted in ALDH assay buffer at 37 °C for 20 min. The ALDH-positive cells were examined by an Accuri C6 cytometer (BD, San Jose, CA, USA).

### 2.11. Quantitative Measurement of Human Cytokines

Human cytokines were measured using the BD^TM^ Cytometric Bead array (CBA) human inflammatory cytokine assay kit (BD, San Jose, CA, USA). MDA-MB-231 cell mammospheres were seeded in ultralow attachment 6-well plates containing 2 mL of a complete MammoCult™ medium for five days and incubated with machilin D (50 µM) for two days. We followed the manufacturer’s protocol. The IL-6 and IL-8 levels in the cultured media were assayed by a BD™ CBA human inflammatory cytokine assay kit. Fifty microliters of mixed capture beads, 50 µL of cultured or standard medium and 50 µL of PE detection reagent were added to each assay tube. The samples were incubated at room temperature for 3 h, protected from light, washed with a washing buffer and centrifuged. After the washes, 300 µL of washing buffer was used to resuspend the pellet, and the samples were analyzed using flow cytometry.

### 2.12. Gene Expression Analysis

Total RNA of the cancer cells was extracted and purified, and real-time RT-quantitative PCR was assayed using a real-time one-step RT-qPCR kit (Enzynomics, Daejeon, Korea). We used a previously described method [30]. The specific primers are described in Appendix A.

### 2.13. Immunofluorescence (IF) Staining Assay

Breast cancer cells were fixed with 4% paraformaldehyde for 30 min, permeabilized with 0.5% Triton X-100 for 10 min, blocked with 3% bovine serum albumin (BSA) for 30 min, and stained with mouse anti-p65, LF-MA30327 (AbFrontier, Seoul, Korea), followed by secondary anti-mouse Alexa 488 antibody, A32723 (ThermoFisher, Walthan, MA, US). We used nonspecific signal conditions to ensure the specificity of the primary antibodies for IF. Finally, the nuclei were stained with DAPI, and p65 was visualized with a fluorescence microscope (Lionheart, Biotek, VT, USA).

### 2.14. Western Blot Analysis

Cells and mammospheres were lysed in ice in a RIPA buffer (1% NP-40, 150 mM NaCl, 1% sodium deoxycholate, 0.1 % SDS, 25 mM Tris-HCl pH 7.6). The lysates were centrifuged and the cytoplasmic extract was collected in the resulting supernatant. Samples (20 μg/10 μL) were prepared from the cells and mammospheres. After electrophoresis on a 12% SDS-PAGE gel, the proteins were transferred to a polyvinylidene fluoride (PVDF) membrane (Millipore, Burlington, MA, USA). The membrane was incubated in an Odyssey blocking buffer at room temperature for 1 h and then incubated overnight with primary antibodies. The antibodies were p65, LF-MA30327; GAPDH, LF-PA0018; Oct4, LF-MA30482 (AbFrontier, Seoul, Korea); LaminB, sc-365962; Nanog, sc-293121; Sox2, sc-365923 (Santa Cruz Biotechnology, Dallas, TX, USA); and c-Myc, 551101 (BD, San Jose, CA, USA). After the membranes were washed, they were incubated with IRDye 680RD and 800W secondary antibodies at room temperature for 1 h, and the signals were determined with an Odyssey CLx machine (Li-Cor, Lincoln, NE, USA).

### 2.15. Electrophoretic Mobility Shift Assay (EMSA)

EMSAs were performed with an Odyssey Infrared EMSA kit (Li-Cor, Lincoln, NE, USA) according to the manufacturer’s instructions. The IRD700-labelled strands of the NF-κB oligonucleotide (5′-/5IRD700/AGTTGAGGGGACTTTCCCAGGC-3′ and 5′-/5IRD700/GCCTGGGAAAGTCCCCTCAACT-3′) were annealed. The IRDye 700 NF-κB oligonucleotide was incubated with nuclear extracts in a final volume of 20 µL at room temperature for 30 min. The samples were electrophoresed on a 6% polyacrylamide nondenaturing gel, and the EMSA data were visualized by an Odyssey CLx machine.

### 2.16. Xenograft Transplantation

Twelve female nude mice were injected with two million MDA-MB-231 cells and injected with/without machilin D (10 mg/kg). The tumor volumes were estimated for 30 days using the formula (width^2^ × length)/2. The mouse experiments were performed as described previously [31]. Animal care and animal experiments were performed in accordance with protocols approved by the Institutional Animal Care and Use Committee (IACUC) of Jeju National University. Female nude mice (four weeks old) were purchased from OrientBio (Seoul, South Korea) and kept in mouse facilities for one week.

### 2.17. Statistical Analysis

All data were analyzed with the GraphPad Prism 7.0 software (GraphPad Prism, Inc., San Diego, CA, USA). All data are reported as the mean ± standard deviation (SD). Data were analyzed by using one-way ANOVA. *P*-values less than 0.05 were considered significant.

## 3. Results

### 3.1. Isolation of a BCSC Inhibitor from S. chinensis

A mammosphere formation assay was performed to screen human BCSC inhibitors using the methanol extracts of *S. chinensis*. The mammosphere assay-based isolation protocol is summarized in Figure 1A. The purified sample inhibited the CSCs (Figure 1B). The extracted samples were purified using organic solvent extraction, silica gel, reversed-phase C18 open column (ODS), a Sephadex LH-20 gel, preparatory TLC, and preparatory HPLC. The isolated compounds were analyzed by preparatory HPLC (Figure 1C). The isolated compound of interest was identified as machilin D (Figure 2).

### 3.2. Machilin D Suppresses Growth and Mammosphere Formation

To determine whether machilin D has a potent inhibitory effect on human cancer cells, we first tested the antiproliferative effect on machilin D at various concentrations in MCF-7 and MDA-MB-231 cells. We observed an antiproliferative effect of ≥ 25 µM machilin D after one day of stimulation (Figure 3A,B). To determine whether machilin D can inhibit mammosphere formation, we treated the mammospheres with machilin D. As shown in Figure 3C,D, machilin D not only decreased the sphere number by 90% but also reduced the size of the mammospheres. Treatment with machilin D suppressed migration, invasion, and colony formation (Figure 3E–G). These results showed that machilin D suppresses mammosphere formation and growth, as well as migration, invasion, and colony formation.

### 3.3. Machilin D Suppresses Tumor Growth

As machilin D has antiproliferative effects in breast cancer, we examined whether this compound reduces tumor growth using an in vivo mouse model. The body weights of the control and machilin D-treated mice did not change (Figure 4A). The tumor volume of the machilin D-treated mice was lower than that of the control mice (Figure 4B). The tumor weights of the machilin D-treated mice were significantly lower than those of the control mice (Figure 4C,D). Our results showed that machilin D effectively decreased tumor growth in a mouse model.

### 3.4. Machilin D Decreases CD44^high^/CD24^low^-expressing and ALDH-positive Cancer Cells

CD44^+^/CD24^-^ and ALDH1 expression are markers of BCSCs. The CD44^+^/CD24^-^ population of breast cancer cells was assayed under machilin D treatment. Machilin D reduced the CD44^+^/CD24^-^ cell fraction from 17.8% to 11.2% (Figure 5A). We also tested the killing effects of machilin D on ALDH-positive cancer cells. Machilin D reduced the ALDH-positive cell fraction from 7.8% to 4.2% (Figure 5B). Our results showed that machilin D specifically inhibits mammosphere formation.

### 3.5. Machilin D inhibits p65 Nuclear Translocation in BCSCs

To investigate the cellular mechanism of machilin D in mammospheres, we determined the localization of NF-κB p65 in mammospheres. We observed that the nuclear level of p65 was significantly decreased following machilin D treatment (Figure 6A). Furthermore, an IF assay of p65 in the MDA-MB-231 cells indicated that the levels of nuclear p65 in the machilin D-treated cells were lower than those in the untreated cells (Figure 6B). Caffeic acid phenethyl ester (CAPE), an inhibitor of the NF-κB signaling pathway that blocks the nuclear translocation of p65, was used to evaluate mammosphere formation [32]. Treatment with machilin D and CAPE (Figure 6C,D), which reduced nuclear p65, blocked mammosphere formation. In conclusion, our data showed that NF-κB signaling regulates mammosphere formation.

### 3.6. Machilin D Reduces the Secretion of IL-6 and IL-8 in the Mammospheres and Regulated NF-κB Activity

To determine the cellular mechanism of machilin D, we have analyzed the localization of NF-κB p65 and secretion of IL-6 and IL-8 of the mammospheres treated with machilin D. NF-κB DNA binding was assessed in the nuclear extracts treated with machilin D by an IRDye 700-NF-κB probe. Machilin D decreased the binding ability of the NF-κB probe (Figure 6E, #3). The specificity of the NF-κB probe was demonstrated using 100 × unlabeled self-competitor (Figure 6E, #4) and 100 × mutated-NF-κB competitor (Figure 6E, #5). Machilin D inhibited the binding ability of NF-κB. Secreted IL-6 and IL-8 are important factors in BCSC survival [33,34]. Real-time RT-qPCR was performed to analyze the transcript levels of IL-6 and IL-8 under machilin D treatment. The data showed that machilin D reduces the transcript levels of IL-6 and IL-8 (Figure 6F). We performed a culture medium cytokine profiling of the mammospheres after machilin D treatment to test the levels of the IL-6 and IL-8 cytokines. The cytokine profiling data showed that machilin D reduced the levels of extracellular IL-6 and IL-8 (Figure 6G).

### 3.7. Machilin D Inhibits the Expression of Stem Cell Marker Genes and the Growth of Mammospheres

To determine whether machilin D inhibits stem cell marker genes, we examined the protein levels of these genes. Machilin D treatment decreased the expression of stem cell markers in human breast CSCs (Figure 7A). To confirm that machilin D reduced mammosphere growth, we incubated mammospheres with machilin D and quantified the cancer cells derived from mammospheres. Machilin D induced cell death and caused a substantial decrease in mammosphere growth (Figure 7B). Our data suggested that machilin D inhibits the growth of BCSCs through NF-κB inhibition and reductions in IL-6 and IL-8.

## 4. Discussion

Breast tumors are the most frequently diagnosed malignancy in breast tissue and are a major cause of cancer-related death in women [35]. Although the mortality of breast cancer has been stabilized, the morbidity is increasing, especially in premenopausal women with a poor prognosis and unfavourable molecular subtypes compared with postmenopausal patients older than 40 years [36]. According to the biological characteristics of the tumor, including stage, grade and genetic status, major treatments, such as surgical resection, chemotherapy and radiotherapy, are selected to increase the survival rate of breast cancer patients [37,38]. The limitation of current therapeutic strategies for patients who display metastasis or experience recurrence has been attributed to the existence of CSCs [39,40]. Increasing evidence has shown that BCSCs characterized by high expression of ALDH and a CD44^high^/CD24^low^ phenotype have multiple drug resistance mechanisms [41]. CSCs derived from cancer cell lines are used as potential targets for breast cancer therapies [42].

Recently, several reports have demonstrated the anticancer effects of *S. chinensis* extracts [26,43]. However, the potential of machilin D, a lignin derived from *S. chinensis*, in cancer or CSC treatment has not been investigated. Our results showed that machilin D has potential as an anti-CSC agent. Machilin D inhibited the growth of breast cancer cells and mammospheres (Figure 3). Metastasis is known to cause high rates of recurrence and mortality in cancer patients [44,45]. Machilin D inhibited the migration, invasion, and colony formation of the breast cancer cells (Figure 3). Furthermore, machilin D inhibited tumor growth (Figure 4).

Current chemotherapeutics target CSCs, inducing cancer relapse. Three subtypes of BCSCs—mesenchymal-like CD44^high^/CD24^low^ CSCs, epithelial-like ALDH^+^ CSCs, and ALDH^+^/CD44^high^/CD24^low^ CSCs—have been identified [46]. Machilin D decreased the CD44^high^/CD24^low^ and ALDH-positive populations (Figure 5). NF-κB signaling is involved in the regulation of inflammation, immunity, cell survival, and growth through the transcription of target genes such as cytokines and growth factors [47]. Constitutively activated NF-κB signaling is present in many types of solid tumors and mediates cancer cell growth and metastasis [48]. During CSC growth as multicellular spheres, constitutive NF-κB signaling was shown to be activated along with upregulation of NF-κB-dependent genes [49]. As the NF-κB signal is important for BCSCs and *S. chinensis* extracts showed significant anti-inflammatory effects [1,50], we examined the expression level and localization of the p65 subunit. Machilin D inhibited mammosphere formation through the NF-κB signaling pathway (Figure 6). CAPE, an inhibitor of NF-κB, decreased p65 translocation and inhibited mammosphere formation (Figure 6).

Several cytokines regulate the NF-κB pathway, and NF-κB also controls the expression of various cytokines, particularly IL-6 and IL-8, which are strongly associated with tumor progression and CSC survival [51,52]. Extracellular IL-6 induces malignant features in stem/progenitor cells from ductal breast carcinoma [33]. In the tumor microenvironment, IL-8 overexpression promotes the acquisition of stemness, mesenchymal features, drug resistance, and the recruitment of immune-suppressive cells that facilitate tumor growth [53]. Novel therapeutics aimed at inhibiting IL-8 receptor signaling may halt tumor progression [54]. Machilin D decreased the secretory IL-6/IL-8 level (Figure 6) and inhibited mammosphere growth (Figure 7). Thus, this compound may be an anticancer agent that targets cancer and BCSCs.

## 5. Conclusions

An isolated compound was identified as machilin D by mass spectrometry and NMR. Machilin D inhibited the growth and mammosphere formation of breast cancer cells and inhibited tumor growth in a xenograft mouse model. Machilin D reduced the CD44^high^/CD24^low^ and ALDH1 cell fractions and the levels of Oct4, Nanog, c-Myc, and Sox2. Furthermore, this treatment reduced the nuclear localization of NF-κB p65 and decreased the secretory IL-6 and IL-8 levels in mammospheres. These results suggest that machilin D blocks NF-κB and IL-6 and IL-8 signaling and induces CSC death. IL-6 and IL-8 regulation is important for breast CSC formation, and machilin D may be a potential agent targeting BCSCs.

## Figures and Tables

**Figure 1 biomolecules-10-00245-f001:**
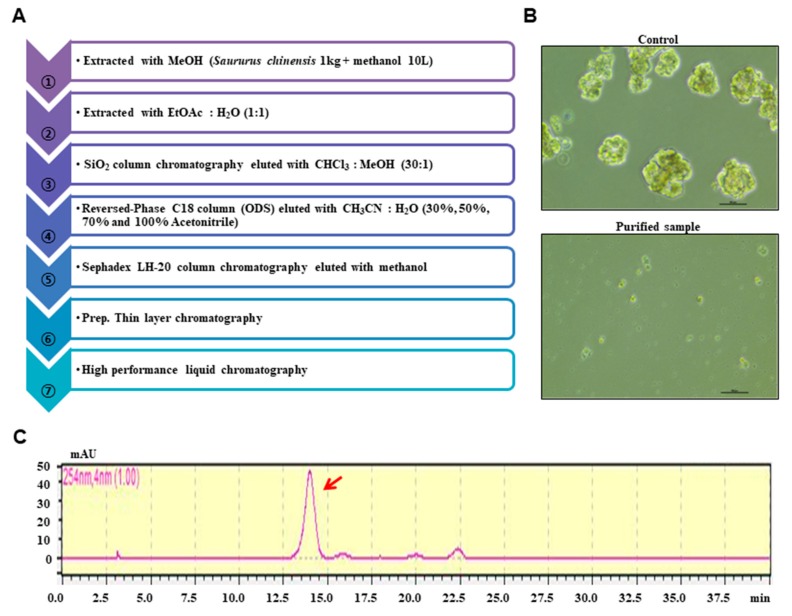
Purification protocol of cancer stem cell (CSC) inhibitors derived from *Saururus chinensis* and mammosphere formation using purified samples. (**A**) Flowchart for the isolation of mammosphere inhibitors. (**B**) Inhibition of mammosphere formation of the high-performance liquid chromatography (HPLC)-purified sample. Cancer cells were treated with HPLC-purified samples. Images show representative mammospheres and were obtained by microscopy (scale bar: 100 μm). (**C**) HPLC analysis of the isolated inhibitor from *S. chinensis*.

**Figure 2 biomolecules-10-00245-f002:**
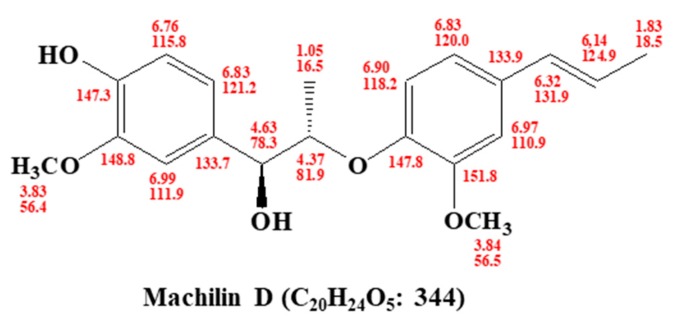
Chemical structure of the isolated compound, machilin D, derived from *Saururus chinensis*.

**Figure 3 biomolecules-10-00245-f003:**
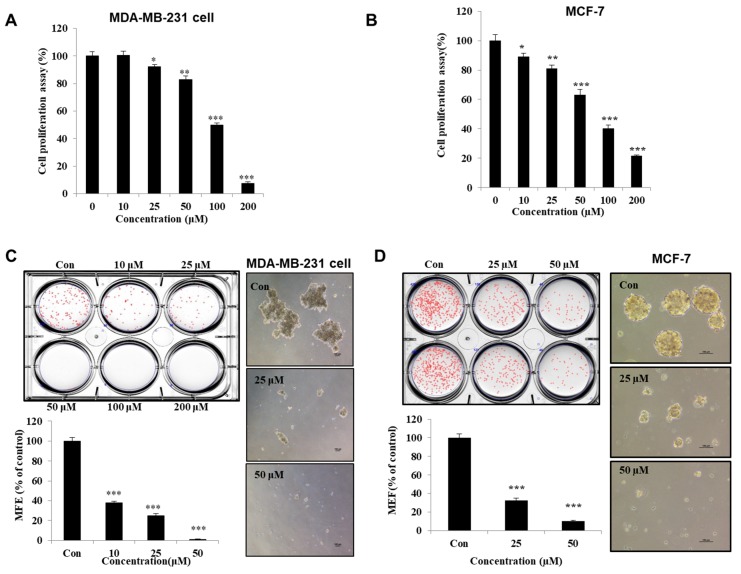
The effect of machilin D on cell growth and mammosphere formation. (**A**) MDA-MB-231 cells were treated with machilin D for 24 h in a culture medium. The cell growth assay using machilin D was measured with an EZ-Cytox kit. (**B**) Breast cancer MCF-7 cells were treated with various concentrations of machilin D for 24 h in a culture medium. The cell proliperation of the MCF-7 cells was measured with an EZ-Cytox kit. (**C**,**D**) Machilin D inhibits the formation of mammospheres. For the establishment of mammospheres, 1 × 10^4^ MDA-MB-231 cells and 4 × 10^4^ MCF-7 cells were seeded in ultralow 6-well plates using a CSC culture media. The mammospheres were incubated with increasing concentrations of machilin D or DMSO for seven days. Images showing representative mammospheres were obtained by microscopy (scale bar: 100 μm). The mammosphere formation efficiency (MFE) was examined. (**E**) Effect of machilin D on the migration of the breast cancer cells. The migration with/without machilin D was photographed at 0 and 24 h (scale bar: 100 μm). (**F**) Transwell assays were performed to determine the cell migration (without Matrigel) and invasion (with Matrigel) of the MDA-MB-231 cells exposed to machilin D (scale bar: 100 μm). (**G**) Machilin D inhibits the colony formation of the cancer cells. The cancer cells were incubated in 6-well plates and treated with machilin D. Representative data were collected. The data from triplicate experiments are represented as the mean ± SD; * *p* < 0.05; ** *p* < 0.01; *** *p* < 0.001.

**Figure 4 biomolecules-10-00245-f004:**
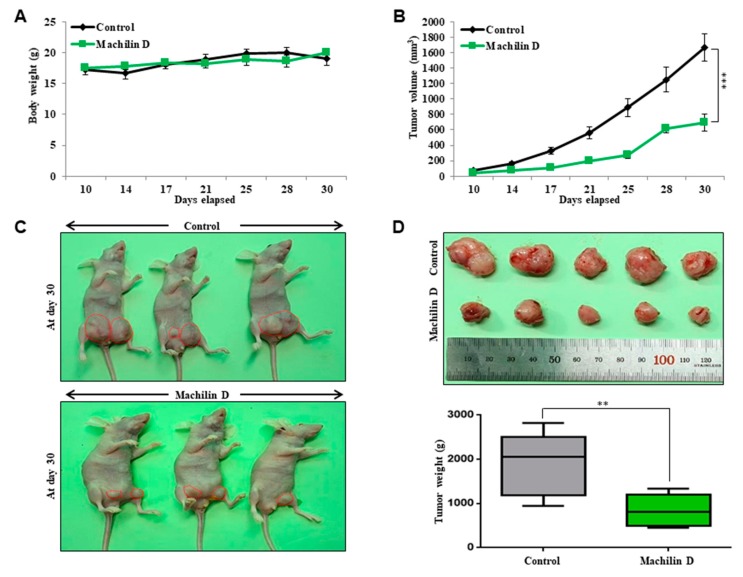
The effect of machilin D on the tumor growth in a xenograft model. MDA-MB-231 cells (2 × 10^6^ cells/mouse) were inoculated into the mammary fat pad of female nude mice and treated with machilin D or DMSO (n = 6). The drug concentration was 10 mg/kg. (**A**) The body weights of the machilin D-treated group were comparable with those of the control group. (**B**) The tumor volume was calculated as (width^2^ × length)/2 at the indicated time points. (**C**) The mouse images of the control and machilin D-treated groups were captured with a camera at day 30. (**D**) Tumor weights of the control and machilin D-treated mice were assayed after sacrifice at day 30. The data are presented as the mean ± SD of three independent experiments. ** *p* < 0.05; *** *p* < 0.01 versus the DMSO-treated control group indicated significant differences.

**Figure 5 biomolecules-10-00245-f005:**
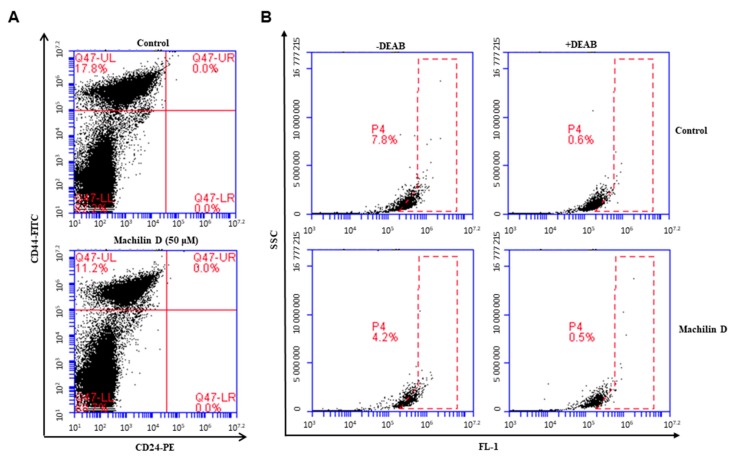
The effect of machilin D on the CD44^high^/CD24^low^ and Aldehyde dehydrogenase (ALDH)-positive cell proportions. (**A**) The CD44^high^/CD24^low^ cell population of the MDA-MB-231 cells treated with machilin D (50 μM) or DMSO for 24 h was analyzed by flow cytometry. The gating was based on binding of a control antibody (red cross). (**B**) Machilin D decreased the ALDH-positive cell population detected by the Aldefluor kit. Cancer cells were treated with machilin D (50 μM) for 24 h and subjected to Fuorescence-activated cell sorting (FACS) analysis. Representative flow cytometric data are shown. The right panel shows the ALDH-positive population with the ALDH inhibitor DEAB, and the left panel represents the ALDH-positive population without DEAB.

**Figure 6 biomolecules-10-00245-f006:**
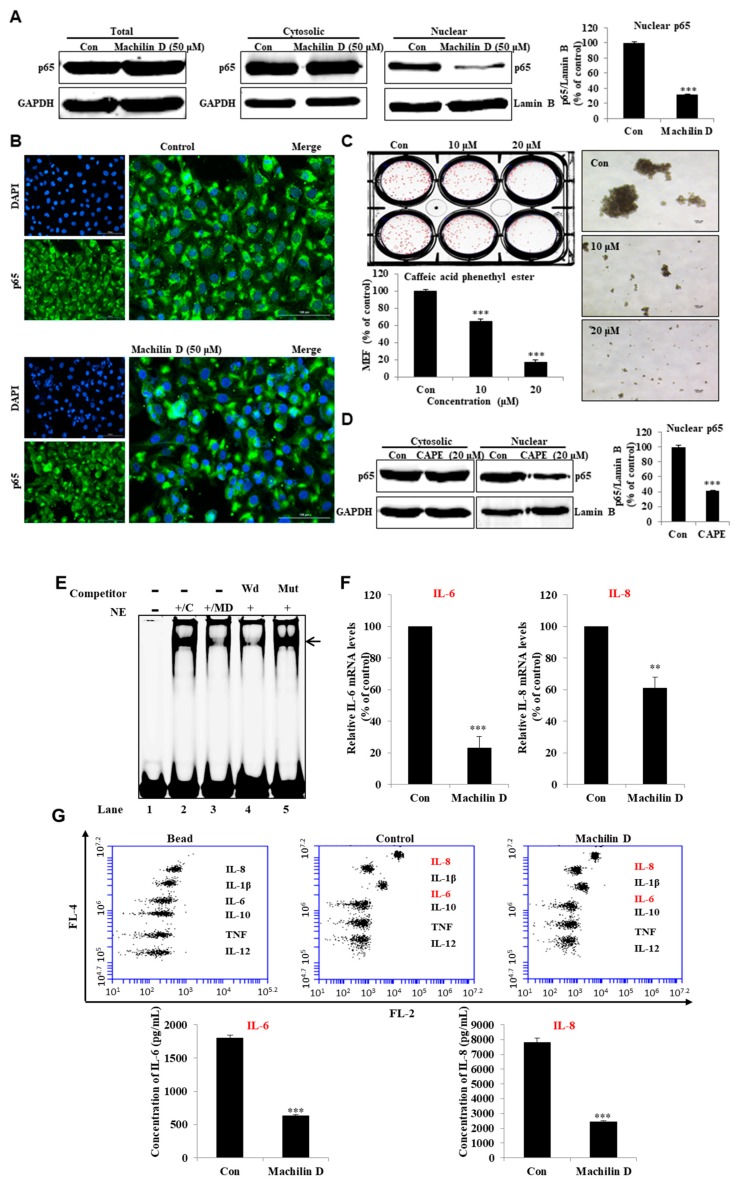
Machilin D regulates the location of NF-κB p65 and secretion of IL-6 and IL-8. (**A**) The levels of p65 in the total, cytosolic, and nuclear proteins were measured in the MDA-MB-231 cells after treatment with machilin D for 24 h using western blot analyses. (**B**) Immunofluorescence (IF) analysis of p65 (green) expression and localization in the breast cancer cells under machilin D treatment. (**C**) The effect of caffeic acid phenethyl ester (CAPE), an inhibitor of NF-κB, on mammosphere formation. (**D**) The p65 levels in the cytosolic and nuclear proteins were measured in the MDA-MB-231 cells after treatment with CAPE for 24 h using western blot analyses. (**E**) Electrophoresis Mobility Shift Assays (EMSAs) of mammosphere nuclear proteins after treatment with machilin D. The nuclear extracts were reacted with the NF-κB probe and were analyzed by 6% native PAGE. Lane 1: NF-κB probe; lane 2: Nuclear extracts with the NF-κB probe; lane 3: Machilin D-treated nuclear proteins with the NF-κB probe; lane 4: Nuclear proteins incubated with the self-competitor (100×) oligo; lane 5: Nuclear extracts incubated with the mutated-NF-κB (100×) probe. The arrow indicates the DNA/NF-κB complex in the mammosphere nuclear lysates. (**F**) Transcriptional expression of the IL-6 and IL-8 genes was determined in the machilin D-treated mammospheres using specific primers. (**G**) Cytokine profile assay of the conditioned media and the machilin D-treated media using specific antibodies and cytokine beads. The data are presented as the mean ± SD of three independent experiments. ** *p* < 0.05; *** *p* < 0.01 versus the DMSO-treated control group indicated significant differences.

**Figure 7 biomolecules-10-00245-f007:**
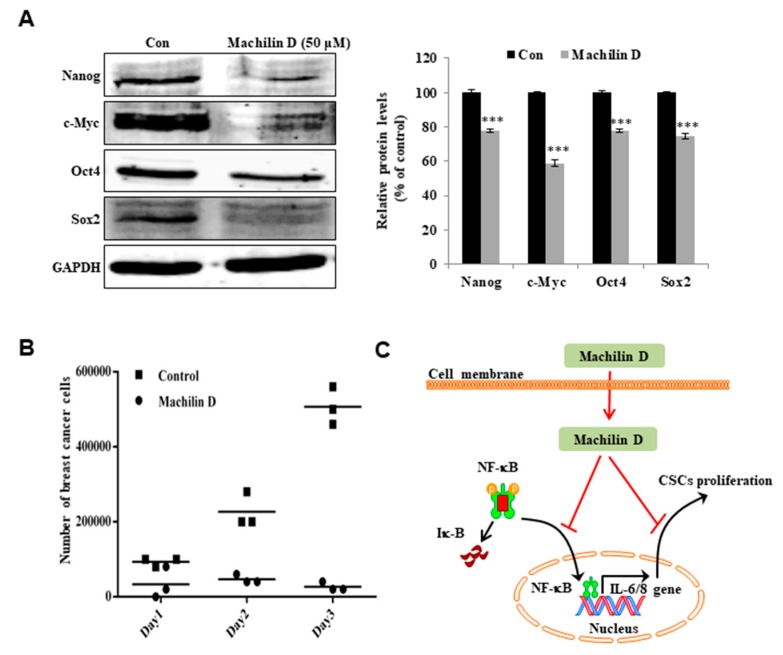
The effect of machilin D on the expression of CSC marker proteins and mammosphere growth. (**A**) Immunoblot analysis of the Nanog, c-Myc, Oct4, and Sox2 proteins in the mammospheres after treatment with machilin D for two days. (**B**) Machilin D inhibits mammosphere growth. Mammospheres with/without machilin D were dissociated into single cells and plated in 6 cm dishes in equal numbers. One, two, and three days later, the cells were counted. (**C**) The proposed model of CSC death through the NF-κB inhibition and IL-6 and IL-8 signaling pathways by machilin D is shown. The data from triplicate experiments are represented as the mean ± SD. ****p* < 0.05 versus the DMSO-treated control group indicated significant differences.

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
