# Peer review of "Machilin D, a Lignin Derived from Saururus chinensis, Suppresses Breast Cancer Stem Cells and Inhibits NF-κB Signaling"

_biomolecules, 2020, doi:10.3390/biom10020245_

Round 1
Reviewer 1 Report
The data presented by Zhen et al. show that machilin D is a tumorsuppressive substance. By using MDA-MB-231 cells, the authors demonstrate that machilin D acts on cell growth, migration and on the cancer stem cell population. Mechanistically, machilin D suppresses NFkappaB activity and inhibits the production of IL-6 and IL-8. These data are convincing. However, the authors claim that the suppression of the NFkappaB activity is the reason for the loss of IL-6 and IL-8 synthesis, but don’t show this for their experimental system. This needs to be demonstrated, also because they state in the title “..NFkappaB-mediated reduction in interleukin-6 and interleukin-8 secretion”. Another weakness of their manuscript is that they used only one breast cancer cell line for their study. The MDA-MB-231 cell line is a frequently used cell line and certainly represents the triple-negative subtype, but it also belongs to a rare subgroup of the TNBCs. Hence, the author should at least show for 1 or 2 additional TNBC lines that machilin D suppresses cell growth and inhibits stem cell activity.
Further comments are given below
lanes 147/148: Why do the author claim that the viability test they are using determines proliferation at 10% FBS. The colorimetric assay measures viable cells, hence it measures growth, not proliferation. Please change accordingly, also in the figures.
Fig. 7: The specificity of the antibodies detecting the stem cell markers Oct4, Sox2 and Nanog should be shown by RNA interference or antibody-blocking peptides. How was the relative protein levels measured? How many Western blots were run for these measurements?
MDA-MB-231 cells use interleukin-6 to keep the activity of STAT3 high. Do the authors see reduced P-STAT3 levels in response to machilin D?
Author Response
I will submit reviewer 1 comments.

Reviewer 2 Report
The manuscript entitled "Machilin D, a Lignin Derived from Saururus chinensis, Suppresses Breast Cancer Stem Cells through NF-κB-mediated Reductions in Interleukin 6 and Interleukin 8 Secretion" by Xing Zhen, Hack Sun Choi, Ji-Hyang Kim, Su-Lim Kim, Ren Liu, Bong-Sik Yun, and Dong‐Sun Lee describes the effect of S. chinensis-derived machilin D on breast cancer stem cells. This study indicates machilin D as a possible therapeutic agent.
Major comments:
The title suggests the direct link between NF-κB and IL-6 and IL-8 inhibition by machilin D. Secretion of IL-6 and IL-8 may not be only NF-κB-dependent. Lines 69-70 indicate inhibition of BCSC via IL-6 and IL-8 regulation. These statements should be presented as assumptions. I would suggest changing the title (and for the title of chapter 3.6, lines 350-351, etc.) to avoid the phrase "NF-κB-mediated" throughout the manuscript. Figure 6 description: what are the control conditions for the experiments?
Minor comments:
Line 45: oestrogen ---> estrogen
Chapter 2.13: Please provide antibody manufacturer
Chapter 2.14: Please provide protein extraction method.
Line 304: NF-kB and p65 ---> NF-κB p65
Line 407: localization of the NF-κB protein --> localization of NF-κB p65
Overall, the manuscript needs certain modifications before it can be published.
Author Response
I will submit reviewer 2 comments.

Round 2
Reviewer 1 Report
Though the authors responded to my comment that the colorimetric method they used measured growth not proliferation by saying:
"We had a mistake and changed material methods and Figure 3 legend".
it still reads "proliferation", not "growth" in the figure, figure legend and the text. Please do the changes. Thanks.
Otherwise, the authors responded satisfactorily to the issues raised.
Author Response
Review 1 comments
Though the authors responded to my comment that the colorimetric method they used measured growth not proliferation by saying:
"We had a mistake and changed material methods and Figure 3 legend".
it still reads "proliferation", not "growth" in the figure, figure legend and the text. Please do the changes. Thanks.
Otherwise, the authors responded satisfactorily to the issues raised.
→The reviewer’s point was well taken.
We changed proliferation in the figure, figure legend and the text into growth.
Reviewer 2 Report
The manuscript entitled "Machilin D, a Lignin Derived from Saururus chinensis, Suppresses Breast Cancer Stem Cells through NF-κB-mediated Reductions in Interleukin 6 and Interleukin 8 Secretion" by Xing Zhen, Hack Sun Choi, Ji-Hyang Kim, Su-Lim Kim, Ren Liu, Bong-Sik Yun, and Dong‐Sun Lee describes the effect of S. chinensis-derived machilin D on breast cancer stem cells. This study indicates machilin D as a possible therapeutic agent.
Minor comments:
The current title: "Machilin D, a Lignin Derived from Saururus chinensis, Suppresses Breast Cancer Stem Cells through Reductions in Interleukin 6 and Interleukin 8 Secretion" should be modified. Although the original title: Machilin D, a Lignin Derived from Saururus chinensis, Suppresses Breast Cancer Stem Cells through NF-κB-mediated Reductions in Interleukin 6 and Interleukin 8 Secretion" has been changed, it still needs modification to avoid the conclusions that machilin D acts through IL-6 and IL-8. I suggest changing the title to: "Machilin D, a Lignin Derived from Saururus chinensis, Suppresses Breast Cancer Stem Cells and inhibits NF-κB signaling". Line 308: NF-kB ---> NF-κB Line 316: please rewrite the sentence (wrong tenses?) Line 323: analyses ---> analyse Line 354: showed ---> suggested?Overall, the manuscript needs few modifications before it can be published.
Author Response
Review 2.
The manuscript entitled "Machilin D, a Lignin Derived from Saururus chinensis, Suppresses Breast Cancer Stem Cells through NF-κB-mediated Reductions in Interleukin 6 and Interleukin 8 Secretion" by Xing Zhen, Hack Sun Choi, Ji-Hyang Kim, Su-Lim Kim, Ren Liu, Bong-Sik Yun, and Dong‐Sun Lee describes the effect of S. chinensis-derived machilin D on breast cancer stem cells. This study indicates machilin D as a possible therapeutic agent.
Minor comments:
The current title: "Machilin D, a Lignin Derived from Saururus chinensis, Suppresses Breast Cancer Stem Cells through Reductions in Interleukin 6 and Interleukin 8 Secretion" should be modified. Although the original title: Machilin D, a Lignin Derived from Saururus chinensis, Suppresses Breast Cancer Stem Cells through NF-κB-mediated Reductions in Interleukin 6 and Interleukin 8 Secretion" has been changed, it still needs modification to avoid the conclusions that machilin D acts through IL-6 and IL-8.
I suggest changing the title to: "Machilin D, a Lignin Derived from Saururus chinensis, Suppresses Breast Cancer Stem Cells and inhibits NF-κB signaling".
→The reviewer’s point was well taken.
We changed the title to: "Machilin D, a Lignin Derived from Saururus chinensis, Suppresses Breast Cancer Stem Cells and Inhibits NF-κB Signaling".
Line 308: NF-kB ---> NF-κB
→ we changed “NF-kB” into “NF-κB”.
Line 316: please rewrite the sentence (wrong tenses?)
→ We added new sentences at Line 316 as followed,
To determine the cellular mechanism of machilin D, we have analyzed the localization of NF-κB p65 and secretion of IL-6 and IL-8 of the mammospheres treated with machilin D
Line 323: analyses ---> analyse
→ We changed “analyses” into “analyse”.
.
Line 354: showed ---> suggested?
→We changed “showed” into “suggested”.
Overall, the manuscript needs few modifications before it can be published.
→The reviewer’s point was well taken. We changed the manuscript.